# Study of a Novel Fluorine-Containing Polyether Waterborne Polyurethane with POSS as a Cross-Linking Agent

**DOI:** 10.3390/polym15081936

**Published:** 2023-04-19

**Authors:** Yajun Deng, Changan Zhang, Tao Zhang, Bo Wu, Yanmei Zhang, Jianhua Wu

**Affiliations:** 1Xiamen Key Laboratory of Marine Corrosion and Intelligent Protection Materials, Jimei University, Xiamen 361021, China; 2Research Center of Graphic Communication, Printing and Packaging, Wuhan University, Wuhan 430079, China

**Keywords:** waterborne polyurethanes, fluorine-containing polyether, POSS, hydrophobicity, mechanical properties

## Abstract

Waterborne polyurethane are more eco-friendly materials due to lower volatile organic compounds (VOCs, mainly isocyanates) content than the alternative materials. However, these rich hydrophilic groups polymers have not yet reached good mechanical properties, durability and hydrophobicity behaviors. Therefore, hydrophobic waterborne polyurethane has become a research hotspot, attracting significant attention. In this work, firstly, a novel fluorine-containing polyether P(FPO/THF) was synthesized by cationic ring-opening polymerization of 2-(2,2,3,3-tetrafluoro-propoxymethyl)-oxirane (FPO) and tetrahydrofuran (THF). Secondly, fluorinated polymer P(FPO/THF), isophorone diisocyanate (IPDI) and hydroxy-terminated polyhedral oligomeric silsesquioxane (POSS-(OH)_8_) were used to prepare a new fluorinated waterborne polyurethane (FWPU). Hydroxy-terminated POSS-(OH)_8_ was used as a cross-linking agent, while dimethylolpropionic acid (DMPA) and triethylamine (TEA) were used as a catalyst. Four kinds of waterborne polyurethanes (FWPU0, FWPU1, FWPU3, FWPU5) were obtained by adding different contents of POSS-(OH)_8_ (0%, 1%, 3%, 5%). The structures of the monomers and polymers were verified by ^1^H NMR and FT-IR, and the thermal stabilities of various waterborne polyurethanes were analyzed by thermogravimetric analyzer (TGA) and differential scanning calorimetry (DSC). As the results, the thermal analysis showed that the FWPU performed the good thermal stability and the glass transition temperature could reach at about −50 °C. The FWPU1 film exhibited that the elongation at break was 594.4 ± 3.6% and the tensile strength at break was 13.4 ± 0.7 MPa, elucidating that the FWPU1 film developed the excellent mechanical properties relative to the alternative FWPUs. Further, the FWPU5 film performed the promising properties, including the higher surface roughness of FWPU5 film (8.41 nm) obtained by the atomic force microscope (AFM) analysis, and the higher value of water contact angle (WCA) at 104.3 ± 2.7°. Those results illustrated that the novel POSS-based waterborne polyurethane FWPU containing a fluorine element could develop the excellent hydrophobicity and mechanical properties.

## 1. Introduction

Waterborne polyurethane has a wide range of practical value in packaging, inks and coatings due to its good mechanical properties and better properties, including wear resistance, flexibility and electrical insulation [1,2,3,4,5]. In addition, owing to using the water as solvent, waterborne polyurethane (WPU) has been widely concerned in replacing traditional organic phase polyurethane, meeting the requirements of low volatile organic compounds (VOCs) [6,7,8]. Due to the existence of a large number of hydrophilic units in WPU, the hydrophobicity and mechanical properties of waterborne polyurethane are relatively poor and limit its wide applications. Therefore, enhancing the hydrophobicity and mechanical properties for the waterborne polyurethane is the meaningful point attracted by the researchers for studying the environmentally friendly and outstanding waterborne polyurethane to expand their application prospects [9,10].

In recent years, numerous studies have shown that fluorine-containing materials revealed excellent properties such as high hydrophobicity, low surface energy and non-viscous energy. Herein, a concept for adding the fluoropolymer to synthesize the WPU is a good strategy for improving the hydrophobicity property of WPU [11,12,13,14,15,16]. In order to enhance the water resistance of WPU, fluorinated polyethylene glycol and fluorinated chain extender are commonly used to modify WPU. For example, Wen et al. reported the synthesis way for a new type of fluorine-containing WPU. The fluorine-containing chains were grafted with different lengths on the main chain, and finally obtained fluorine-containing WPU. The results showed that fluorocarbon chain could improve the hydrophobicity, thermal stability and mechanical properties of waterborne polyurethane [17]. Sui et al. synthesized a series of fluorine-containing waterborne polyurethane emulsions (H-SiFPU) with hydroxypropyl polydimethylsiloxane (HP-PDMS) and polytetramethylene ether diol (PTMG). The results showed that with the increased amounts of HP-PDMS, the particle size firstly increased and then decreased, accompanied by the gradually decreased transmittance. Moreover, the addition of HP-PDMS could prospectively enhance the meaningful properties, including thermal stability, aging resistance, and the hydrophobicity [18].

Silicon-based material is another hot material that has been widely used in the modification of polyurethane materials because of its low surface energy, good biocompatibility, high thermal and oxidation stability. It is one of the ideal methods to enhance the hydrophobicity of WPU. The polyhedral oligomeric silsesquioxane (POSS) depends on the special structure to be the optimal choice. POSS is an inorganic and organic component with a rigid structure [19,20] which is associated with the chemical and physical bonds to form the synergistic effect to efficiently perform the thermal, mechanical and surface properties. Thus, it becomes the most potential representative in the process of the material structure design [21,22,23,24,25]. Ajaya K. Nanda et al. [26] studied the effect of diamino-POSS on the properties of polyurethane materials. The tensile strength and glass transition temperature (T_g_) were significantly improved by the experimental results. A POSS-based material was synthesized from dihydroxy and two triethoxy groups by Zhao et al. [27], and it was found that adding 8 wt% m-POSS into the polymer could exhibit the good properties including high hydrophobicity (121°), tensile strength (15.3 MPa) and the increasing elongation at break (1000%). Chen et al. investigated a feasible common end sealing strategy for the synthesis of monofunctional POSS to modify waterborne polyurethane (WPU). Two modifiers including monofunctional aminopropylisobutyl-POSS (AIPOSS) and (3-aminopropyl) triethoxysilane (APTES), were used to seal PU chains. The POSS in silane with different additions were used to modify PU to significantly enhance the mechanical properties of the films. The formation of cross-linking network highly enhanced due to the additional physical cross-linking points on chemical substances. Besides, the surface hydrophobicity and dielectric properties were promoted by the establishment of POSS. In addition, a membrane was formed after treatment with a fluorosilane coupling agent. Fluorine POSS–PU hybrid film exhibited significant hydrophobicity and low dielectric constant [28]. Therefore, it is a very interesting study to improve the performance of waterborne polyurethane films via modifying POSS as the cross-linking agent in the waterborne polyurethane system.

In this work, a novel fluorine-containing polyether P(FPO/THF) was synthesized by cationic ring-opening polymerization of 2-(2,2,3,3-tetrafluoro-propoxymethyl)-oxirane (FPO) and tetrahydrofuran (THF) [29,30]. The main chain structure of the polyether structure showed the lower glass transition temperature [31]. Then, fluorinated polymer P(FPO/THF), isophorone diisocyanate (IPDI) and hydroxy-terminated POSS-(OH)_8_ were used to prepare a new fluorinated waterborne polyurethane (FWPU). Hydroxy-terminated POSS-(OH)_8_ was used as a cross-linking agent and dimethylolpropionic acid (DMPA) and triethylamine (TEA) were used as a catalyst. Four kinds of waterborne polyurethanes (FWPU0, FWPU1, FWPU3, FWPU5) were obtained by adding different contents of POSS-(OH)_8_ (0%, 1%, 3%, 5%). The monomers and polymer structures were verified by ^1^H NMR and FT-IR, and we investigated the hydrophobicity, thermal stability, mechanical properties and surface morphology of the environmentally friendly POSS-based FWPU films by water contact angle (WCA), thermogravimetric analyzer (TG), differential scanning calorimetry (DSC), scanning electron microscopy (SEM) and atomic force microscope (AFM).

The results of the study showed that we reported the method to improve the poor hydrophobicity and weak mechanical properties of waterborne polyurethanes with POSS and fluorinated materials. Firstly, a new fluorinated polyether was synthesized to demand the needs which contained the low surface energy property of fluorine element and the low glass transition temperature property of polyether. Secondly, based on the good hydrophobicity of POSS, it becomes a cross-linking agent for waterborne polyurethane with the modification of POSS. Finally, a new waterborne polyurethane was synthesized from the fluorinated polyether and the modified POSS. We tested the new waterborne polyurethane film, and found that its hydrophobicity and mechanical properties had reached the satisfy results with the greatly improved values. The FWPU1 film showed that the elongation at break was 594.4 ± 3.6% and the tensile strength at break was 13.4 ± 0.7 MPa, elucidating that the FWPU1 film developed the excellent mechanical properties relative to the alternative FWPUs. In addition, the FWPU5 film performed the promising properties, including the higher surface roughness of FWPU5 film (8.41 nm) obtained by atomic force microscope (AFM) analysis, and the higher value of water contact angle (WCA) at 104.3 ± 2.7°. The above results illustrated that the proper contents of POSS in the WPU system could enhance the properties to meet the demands and expand their applications.

## 2. Experiment

### 2.1. Materials

POSS-octavinyl, 2,2-azobisisobutyronitrile (AIBN), isophorone diisocyanate (IPDI), dimethylolpropionic acid (DMPA), glycidyl 2-(2,2,3,3-tetrafluoro-propoxymethyl)-oxirane (FPO) and 2-hydroxy-1-ethanethiol were purchased from Aladdin (Shanghai) Co., Ltd. (Shanghai, China). Triethylamine (TEA), 1,4-butanediol (BDO), dibutyltin dilaurate (DBTDL), acetone and dichloromethane were available from Sinopharm Chemical Reagent Co., Ltd. (Shanghai, China). Tetrahydrofuran (THF, 99.5%, superdry with molecular sieves) and boron trifluoride diethyl ether were purchased from J&K scientific Ltd. (Shanghai, China). Acetone was stored with 4 Å type molecular sieves for further use. Other reagents and solvents were utilized without any purification.

### 2.2. Synthesis of Hydroxyl Terminated POSS-(OH)_8_

Hydroxyl terminated POSS-(OH)_8_ was synthesized by thiolene click method. POSS-octavinyl and 2-hydroxy-1-ethanethiol were reacted with the tetrahydrofuran solution at 60 °C for 24 h under nitrogen atmosphere, and AIBN was used as the initiator. After cooling to room temperature, the crude product was precipitated several times to obtain the product POSS-(OH)_8_. The protocol for the synthesis process of the hydroxyl-terminated POSS-(OH)_8_ was illustrated in Figure 1.

### 2.3. Synthesis of Fluorine-Containing Hydroxyl Terminated Polyether P(FPO/THF)

The synthesis route of P(FPO/THF) was shown in Figure 1. 1,4-BDO as initiator and BF_3_·OEt_2_ as catalyst were added into dry CH_2_Cl_2_ (5 mL) in the flask and stirred at 0–5 °C for 1 h. Then, FPO and THF were dissolved in dry CH_2_Cl_2_ (10 mL) which was added dropwise into the above flask. The reaction mixture was stirred and kept at 0 °C for 24 h. The reaction was terminated by water and then the mixture was adjusted to neutral pH value by water. After separated, the organic layer was washed with water several times and then dried under vacuum to obtain P(FPO/THF) polymer (Yield: 89.3%, M_n_ = 2362 g/moL, PDI = 1.58). The synthesis process of the P(FPO/THF) was depicted in Figure 1.

### 2.4. Preparation of POSS-Based FWPU Dispersions

The formulations of POSS bonded for the preparation POSS-based WFPU dispersions, as shown in Table 1. Firstly, IPDI was put into a three-neck round bottom flask for heating at 65 °C and stirred under nitrogen atmosphere. P(FPO/THF) was dissolved in acetone (10 mL), and then slowly dropped into the flask at 85 °C for 30 min. DMPA was added to the flask above mentioned, and the reaction was continuously heated for 3 h. Afterwards, POSS-(OH)_8_ as a cross-linking agent was introduced into the reaction at the different proportions (0 wt%, 1 wt%, 3 wt%, and 5 wt%). Then, the DBTDL (as a catalyst for 1% of the total mass) was added into the flask to react for 2 h under stirring condition, and the certain amount of acetone was utilized to reduce the viscosity of the reacted mixture. When the mixture was cooled down at 35 °C, TEA, with the same ratio as DMPA, was added to neutralize the carboxyl group of DMPA. The deionized water was used to terminate the reaction under vigorous stirring of 1100 rpm for 15 min. Finally, the mixture was removed from acetone for 3 h under 0.1 MPa vacuum, and the FWPU dispersion was obtained with 35 wt% of solid content. The synthesis process of the POSS-based FWPU was depicted in Figure 2.

### 2.5. Preparation of POSS-Based FWPU Films

The films were obtained by pouring the FWPU dispersions into glass molds at the room temperature for 3 d, and then the FWPU films further dried at 40 °C in a vacuum drying oven for 2 d to completely remove the residual water. The vacuum dried films were stored in a desiccator to avoid moisture.

### 2.6. Characterizations

^1^H NMR (Nuclear Magnetic Resonance) spectra measurement was recorded on a spectrometer (Mercury VX-300, Varian, Palo Alto, CA, USA). The sample of POSS-(OH)_8_ was dissolved in DMSO-d6 and the sample of P(FPO/THF) was dissolved in CDCl_3_ containing tetramethyl silane as internal standard and packed in the nuclear magnetic tubes.

FT−IR (Fourier Transform Infrared Spectroscopy) of the samples in attenuated total reflection (ATR) infrared mode was obtained on a Thermo Scientific Nicolet, (iS50, Waltham, MA, USA) and the spectral range was varied from 4000 to 500 cm^−1^.

The thermal stability was performed on a thermogravimetric measurement (TG209 F1 Libra thermobalance, Netzsch, Germany). About 5–10 mg samples were heated from 25 to 600 °C with a heating rate of 10 °C/min under a nitrogen atmosphere (20 mL/min).

The thermal analysis of four polymers was carried out using the DSC analyzer (Q100, New Castle, DE, USA). About 5 mg samples were sealed in aluminum crucibles and heated from −60 to 200 °C at a heating rate of 10 °C/min under nitrogen (20 mL/min) condition.

The mechanical property measurements were performed with a universal tensile testing machine (Instron 3343, Instron Corp, Norwood, MA, USA) at a crosshead speed of 100 mm/min. Dumbbell-shaped specimens were cut to form 1 mm thick films with a gauge section 25 mm × 5 mm.

Water contact angle (WCA) value of different FWPU films was recorded with a video optical contact angle measuring instrument (OCA15EC, DataPhysics Instruments GmbH, Filderstadt, Germany) via sessile method, and deionized water (4 μL) was dropped slowly on the film surface. WCA value of each sample was measured after 5 s.

The morphology of FWPU films was examined by field emission scanning electron microscopy (Hitachi FE−SEM 4800, Tokyo, Japan) under an acceleration voltage of 15 kV, and the surface roughness of FWPU films was recorded by atomic force microscope (AFM, Dimension Icon, Bruker, Billerica, MA, USA).

## 3. Results and Discussion

### 3.1. H NMR and FT−IR Analysis of the Monomer and Different Polymers

In order to verify the successful synthesis of fluorine-containing waterborne polyurethane FWPU, the structure of the product was monitored with FT−IR and ^1^H NMR technology. The ^1^H NMR spectra of POSS-(OH)_8_ and P(FPO/THF) are shown in Figure 1. As shown in Figure 1a, the feature peak at 4.71 ppm belonged to the protons of the end hydroxyl group (–OH). The proton of –CH_2_OH was observed at 3.71 ppm [32]. The peaks at 2.47–2.60 ppm were ascribed to the –CH_2_SCH_2_– unit. The –SiCH_2_– unit was identified by the peak at 0.94 ppm. These characteristic peaks indicated that POSS-(OH)_8_ were successfully synthesized. As shown in Figure 1b, the peaks at 5.71–6.11 ppm were ascribed to the –CF_2_H group [33]. The peaks at 1.61 ppm and 3.78–4.05 ppm corresponded to the –OCH_2_CH_2_CH_2_CH_2_O– and –OCH_2_CF_2_–. It was illustrated that POSS-(OH)_8_ and P(FPO/THF) were successfully synthesized by ^1^H NMR technology.

The different contents of POSS-(OH)_8_ were introduced into the FWPU system and those composite films were analyzed by FT−IR, as shown in Figure 2. The peaks at 3318 cm^−1^ and 1701 cm^−1^ belonged to the stretching vibrations of the N–H and C=O units. The stretching vibration movement of the C–H bond in the CH_2_ and CH_3_ groups was attributed to the adsorption bands at 2843–2952 cm^−1^. The band at 872 cm^−1^ was ascribed to the Si–O–Si asymmetric [34]. The characteristic signals at 1234 cm^−1^ and 1535 cm^−1^, respectively, corresponded to the C–O–C bond and the bending of the C(O)–NH bond. The rocking vibration of C–F bond was identified by the bands at 824 cm^−1^ and 692 cm^−1^, concretely, and the 1098 cm^−1^ band was defined as the C–F and C–O–C bonds. It was identified by the characterization from the FT−IR measurement that the prospective structure of the various FWPU films were successfully fabricated.

### 3.2. Thermal Analysis of the Different Films

The thermal decomposition of diverse composite films (FWPU0, FWPU1, FWPU3 and FWPU5) was measured by TGA and DTG analyzers. As shown in Figure 3a, the four films decomposed at plural between 200–450 °C, and the decomposition of all samples ends at 450 °C. As shown in Figure 3b, it was observed that the thermal decomposition of FWPU had three stages. It was the first stage degradation of the POSS chain at the range from 271.5 °C to 273.5 °C. Because of the urethane bond, the second decomposition stage was formed at 338 °C, and the final stage was the maximum breakage at 398 °C owing to the decomposition of polyether chain. In addition, to further analyze the thermal properties and phase structure of the FWPU films, the glass transition temperature (T_g_) and melting temperature (T_m_) of waterborne polyurethane films were carried out by the DSC analyzer. The polyether structure, as the soft chain segment in the FWPU network, induced the T_g_ value of all the FWPU films at about −50 °C, as shown in Figure 3c. In addition, in the composite FWPU films, the more contents of POSS added, the higher T_m_ values increased. For example, the FWPU0 film showed the lowest T_m_ at 92.6 °C, while the FWPU5 exhibited the highest T_m_ at 112.3 °C. The results showed that the melting temperature of the hard segment improved with the increase in the content of the POSS cross-linking agent [35]. This was because the POSS cross-linking of individual main chain structures reduced the molecular mobility in waterborne polyurethane molecular.

### 3.3. Water Contact Angle (WCA) and Mechanical Properties of the Different Films

The water contact angle (WCA) directly illustrates the surface character about the hydrophilic and hydrophobic features of the films. Herein, the various WPU films, including FWPU0, FWPU1, FWPU3 and FWPU5, were measured by CA analyzer, and results from this measurement were developed the satisfactory consequence in Figure 4a–d. The FWPU0 film possessed the CA value at 68.3 ± 1.2° because of the fluorine-based side chain in polyether, and it was obvious higher than pure polyether chain waterborne polyurethane film owing to the hydrophilic groups in the WPU system. The increased contents of the POSS were introduced into the film, the CA values of FWPU1 to FWPU5 were increased gradually from 92.9 ± 1.2° to 104.3 ± 2.7°, as shown in Figure 4b–d. It was because that the fluorine-based soft chain in the FWPU could greatly reduce the surface energy and achieve the hydrophobic effect of the composite films. It also effected on the silicon element of the POSS structure to enhance the hydrophobicity of the FWPU [36,37].

The stress-strain of the films was tested by the tensile measurement to further analyze the mechanical properties in the deformation process of the various films. The stress-strain curves as well as the values of tensile strength and elongation at break were shown in Figure 5 and Table 2, respectively. Relative to the control FWPU0 film, the FWPU1 film performed the good mechanical behavior because the stress had increased from 6.9 ± 0.5 MPa to a maximum of 13.4 ± 0.7 MPa. However, it was not the increase in POSS content that enhanced the stress strength, and it was identified that the strength of the FWPU3 and FWPU5 films, respectively, decreased to 8.6 ± 0.3 MPa and 6.2 ± 0.3 MPa. This was mainly due to the enhanced rigidity with the increase in silicon–oxygen bond content in the POSS structure, which lead to the failure of the network results. Further, the elongation at break had been further studied, and the results elucidated that the composite FWPU films including the FWPU0, FWPU1, FWPU3 and FWPU5 showed the elongation at 290.7 ± 2.8%, 594.4 ± 3.6%, 512.2 ± 2.1% and 410.8 ± 1.3%, respectively. This consequence illustrated that a certain amount of the POSS cross-linking agent could effectively improve the mechanical properties of waterborne polyurethane. It was because the POSS cross-linking agent was the vital parameter to increase the link density in the waterborne polyurethane system. Thus, the interaction for molecules strengthen could lead to significant improvement of the mechanical properties for the composite materials [28,38]. When the content of POSS increased, the rigidity of the molecular chain could enhance, and the mobility of the molecular chain would reduce, resulting in the decrease in mechanical properties.

### 3.4. Morphology Analysis

The surface roughness (5 × 5 μm^2^) of the different films were carried out by AFM, and the images were collected in Figure 6. It was observed that the surface roughness R_a_ value of FWPU0 (Figure 6a) without any POSS was 2.49 nm. However, with the addition of POSS nanoparticles at the rate of 1%, 3% and 5%, the surface roughness of the waterborne polyurethane films of FWPU1 (Figure 6b), FWPU3 (Figure 6c) and FWPU5 (Figure 6d) increased to 3.24 nm, 6.47 nm and 8.41 nm, respectively. Since POSS mainly consists of nanomaterial structure composed of silicon and oxygen elements, the roughness of waterborne polyurethane gradually improved with the increased POSS contents. This structure exhibited that the addition of POSS nanoparticles improved the hydrophobicity of the waterborne polyurethane, which had the same conclusion demonstrated by the WCA.

Deeply surface morphology of the composite films FWPU were studied by SEM measurement, as shown in Figure 7. Without the addition of a POSS cross-linking agent, the waterborne polyurethane FWPU0 (Figure 7a) surface was very regular and a little gully. However, with the addition of POSS nanoparticles, the surface of waterborne polyurethane obviously showed some prominent and small points which were caused by the addition of nanoparticles. Due to the increased amount of nanomaterials, the cross-linking density strengthens excessively, leading to the local aggregation of waterborne polyurethane and the production of some small bumps. When 1% POSS was added, the surface of FWPU1 film (Figure 7b) became more regular and denser owing to the enhanced cross-linking density. With the increase in POSS content to 3% and 5%, the surface of FWPU3 (Figure 7c) and FWPU5 (Figure 7d) films became denser, but the rigidity and irregularity further increased, resulting in the decreased mechanical properties. Therefore, when the appropriate amount of POSS cross-linking agent increased, the stiffness of the molecular chain would increase, but the mobility of the molecular chain would reduce, and a little gully would become regular.

## 4. Conclusions

In order to improve the hydrophobicity and mechanical properties of the ecofriendly waterborne polyurethane, the environmental-friendly materials exhibit the promising trend and extend their applications. In this work, a novel fluorinated polyether P(FPO/THF) was synthesized by cationic ring-opening polymerization with 2-(2,2,3,3-tetrafluoro-propoxymethyl)-oxirane (FPO) and tetrahydrofuran (THF). Then, a novel fluorinated waterborne polyurethane (FWPU) was prepared using fluorinated polymer P(FPO/THF), isophorone diisocyanate (IPDI) and hydroxyl-terminated POSS-(OH)_8_. Four kinds of waterborne polyurethane (FWPU0, FWPU1, FWPU3, FWPU5) were prepared by using hydroxyl-terminated POSS-(OH)_8_ as a cross-linking agent. The monomers and polymer structures were verified by ^1^H NMR and FT-IR, and the thermal stability of various waterborne polyurethanes was analyzed by a thermo-gravimetric analyzer (TGA) and a differential scanning calorimetry (DSC). The thermal analysis showed that the FWPU developed good thermal stability and the glass transition temperature could reach about −50°C, identifying the good mechanical properties even at the low temperature. The mechanical analysis of four kinds of waterborne polyurethane films FWPU found that FWPU1 film had the best mechanical properties (elongation at break was 594.4 ± 3.6%, tensile strength at break was 13.4 ± 0.7MPa). Then, we carried out atomic force microscope (AFM) analysis on the films and found that the surface roughness of FWPU5 film could reach 8.41 nm. Therefore, the water contact angle (WCA) of FWPU5 was tested to be 104.3 ± 2.7°. The diversity of both chemical components and cross-linking strategies can be achieved to make the waterborne polyurethane to be a unique class of materials whose beneficial properties were involved in hydrophobicity and good mechanical performances. The fluorine grafted chain hold the post of the vital components in the polyether section. The nanoparticle network such as POSS as the cross-linking agent can reinforce the mechanics through promoting the fluorine-based polyether to form FWPU. After cooperating with chemical components and cross-linking agent in the FWPU structure, the FWPU films can be used to adjust materials approving mechanical functions such as high tensile strength, and even hydrophobicity.

## Data Availability

The data presented in this study are available on request from the corresponding author.

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
