# Peer review of "Study of a Novel Fluorine-Containing Polyether Waterborne Polyurethane with POSS as a Cross-Linking Agent"

_polymers, 2023, doi:10.3390/polym15081936_

Round 1

Reviewer 1 Report

Please read and “fully” address the comments listed below:

1.              The ABSTRACT is not written in a logical order. Start with an overview of the topic and a rationale for your paper. Describe the methodology you used and the general outline of the manuscript. Also, in the end, state the result in more detail (i.e., provide some numbers).

2.              The novelty of your work is still unclear to the reader, which should be further detailed both in the Abstract and Introduction. In other words, the purpose of the research is missing, which must be clearly presented.

3.              Show POSS contents for each subplot of fig.4.

4.              Fig. 4 shows the static contact angle (CA) measurement on the films, please determine when the CA was measured (e.g., at 0.1 or 0.5 seconds after dispensing the drops?). Also, discuss whether dynamic CA measurements (change of CA with time) are needed to better explore the fluorine-containing polyether waterborne polyurethane with POSS as cross-linking agent

5.              Scale bars are missing from many figures, e.g., fig. 4. 

6.              Provide more explanation for this sentence: Page 9: " However, with the addition of POSS nanoparticles at the rate of 1%, 3% and 5%, the surface roughness of the water-borne polyurethane films of FWPU1 (Fig. 6 (b)), FWPU3 (Fig. 6 (c)) and FWPU5 (Fig. 6 (c)) increased to 3.24 nm, 6.47 nm and 8.41 nm, respectively”

7.              Similarly, this sentence needs to be better explained: Page 10: “However, with the addition of POSS nanoparticles, the surface of waterborne polyurethane obviously showed some prominent small points which were caused by the addition of nanoparticles”.

8.              In Fig. 4 the authors show the CA of drops in the last subplot (i.e., Fig. 4d) which is ~ 104.3 degrees. However, it is not determined how this was measured, was it based on polynomial fitting (by commercial goniometers) or manual measurements? As a result, please write a paragraph in your manuscript that recently deep learning-based methods are developed to accurately measure the contact angle of drops regardless of the skills and experience of the operator, and they do not have the main limitations of polynomial fittings, especially in the presence of optical noises, e.g., diffracting, scattering, and blurring (and cite or reference the two papers listed below)

·      Zhang, Z., & Song, X. (2021). Characterizing the impact of temperature on clay-water contact angle in geomaterials during extreme events by deep learning enhanced method. In Geo-Extreme 2021 (pp. 160-168).

·      Kabir, H., & Garg, N. (2023). Machine learning enabled orthogonal camera goniometry for accurate and robust contact angle measurements. Scientific Reports, 13(1), 1497.

9.     Conclusion: Can authors highlight future research directions and recommendations? Also, highlight the assumptions and limitations (e.g., shortcomings of the present study). Besides, recheck your manuscript and polish it for grammatical mistakes (you can use “Grammarly” or similar software to quickly edit your document).

Author Response

Point-to-point responses to the comments from the reviewers

We would like to express our sincere appreciation for the useful comments and constructive suggestions from the reviewer 1. The resubmitted manuscript has been completely revised according to all the comments. The itemized responses to each the comments are listed as below.

Point 1: The ABSTRACT is not written in a logical order. Start with an overview of the topic and a rationale for your paper. Describe the methodology you used and the general outline of the manuscript. Also, in the end, state the result in more detail (i.e., provide some numbers).

Response 1: Thanks for the comment. I have made a full correction to the abstract in the manuscript.

Point 2: The novelty of your work is still unclear to the reader, which should be further detailed both in the Abstract and Introduction. In other words, the purpose of the research is missing, which must be clearly presented.

Response 2: Thanks for the comment. I have rewritten the abstract and introduction in the manuscript.

Point 3: Show POSS contents for each subplot of fig.4.

Response 3: Thanks for the comment. I have showed POSS contents for each subplot of Fig. 4.

Point 4: Fig. 4 shows the static contact angle (CA) measurement on the films, please determine when the CA was measured (e.g., at 0.1 or 0.5 seconds after dispensing the drops?). Also, discuss whether dynamic CA measurements (change of CA with time) are needed to better explore the fluorine-containing polyether waterborne polyurethane with POSS as crosslinking agent

Response 4: Thanks for the comment. Because I synthesized four kinds of waterborne polyurethanes with different contents of POSS to explore the influence of different contents of POSS on the hydrophobicity and mechanical properties of waterborne polyurethanes. Therefore, I tested the water contact angle when water droplets touched the interface at the same time for this research. Thank you very much again for your advice.

Point 5: Scale bars are missing from many figures, e.g., fig. 4.

Response 5: Thanks for the comment. Sorry, because Figure 4 is a picture format, which is the fixed size value of a water drop. There is no similar scale bar in many literatures. (https://doi.org/10.1016/j.porgcoat.2022.107242 ) Besides, I really don't know how to add scale bar in Fig 4. Thanks again for the comment.

Point 6: Provide more explanation for this sentence: Page 9: " However, with the addition of POSS nanoparticles at the rate of 1%, 3% and 5%, the surface roughness of the water-borne polyurethane films of FWPU1 (Fig. 6 (b)), FWPU3 (Fig. 6 (c)) and FWPU5 (Fig. 6 (c)) increased to 3.24 nm, 6.47 nm and 8.41 nm, respectively”

Response 6: Thanks for the comment. Since POSS mainly consists of nanomaterial structure composed of silicon and oxygen elements, the roughness of waterborne polyurethane gradually increases with the increase of POSS content, which is also shown in the AFM diagram. I have put the explanation in 3.4 Morphological analysis.

Point 7: Similarly, this sentence needs to be better explained: Page 10: “However, with the addition of POSS nanoparticles, the surface of waterborne polyurethane obviously showed some prominent small points which were caused by the addition of nanoparticles”.

Response 7: Thanks for the comment. Due to the increase in the amount of nanomaterials, the crosslinking density increased excessively, leading to the local aggregation of waterborne polyurethane and the production of some small bumps. I have put the explanation in 3.4 Morphological analysis.

Point 8: In Fig. 4 the authors show the CA of drops in the last subplot (i.e., Fig. 4d) which is ~ 104.3 degrees. However, it is not determined how this was measured, was it based on polynomial fitting (by commercial goniometers) or manual measurements? As a result, please write a paragraph in your manuscript that recently deep learning-based methods are developed to accurately measure the contact angle of drops regardless of the skills and experience of the operator, and they do not have the main limitations of polynomial fittings, especially in the presence of optical noises, e.g., diffracting, scattering, and blurring (and cite or reference the two papers listed below).

Response 8: Thanks for the comment. I am very sorry that I forgot to put the instrument measurement method in the manuscript. I have put the water contact Angle measurement method in 2.6 Characterizations. Thank you again for your valuable advice.

Zhang, Z., & Song, X. (2021). Characterizing the impact of temperature on clay-water contact angle in geomaterials during extreme events by deep learning enhanced method. In Geo-Extreme 2021 (pp. 160-168).

Kabir, H., & Garg, N. (2023). Machine learning enabled orthogonal camera goniometry for accurate and robust contact angle measurements. Scientific Reports, 13(1), 1497.

Point 9: Conclusion: Can authors highlight future research directions and recommendations? Also, highlight the assumptions and limitations (e.g., shortcomings of the present study). Besides, recheck your manuscript and polish it for grammatical mistakes (you can use “Grammarly” or similar software to quickly edit your document).

Response 9: Thanks for the comment. I have revised and supplemented the conclusion in the manuscript.

Reviewer 2 Report

The paper of Deng et al.’ Study of a novel fluorine-containing polyether waterborne polyurethane with POSS as cross-linking agent’ deal with the synthesis of PU with fluorine-containing polyether soft blocks.

 First of all the paper should be rewrite because the English is good for nothing.

See abstract: P(FPO/THF) was mainly synthesized by 2,2,3,3-tetrafluoropropyl ether (FPO) and tetrahydrofuran (THF) via cationic open-ring collection. (???)

The used chemical names are also not correct.

See Materials dimethy iolpropionic acid (DMPA)- ???

Results

Page 7 TGA all samples degraded completely at 450. How it possible if the sample contains 5 wt% of POSS Should stay Si containing ash.

In this form the paper should be reject.

Author Response

Point-to-point responses to the comments from the reviewers

We would like to express our sincere appreciation for the useful comments and constructive suggestions from the reviewer 2. The resubmitted manuscript has been completely revised according to all the comments. The itemized responses to each the comments are listed as below.

The paper of Deng et al.’ Study of a novel fluorine-containing polyether waterborne polyurethane with POSS as crosslinking agent’ deal with the synthesis of PU with fluorine-containing polyether soft blocks.

First of all the paper should be rewrite because the English is good for nothing.

See abstract: P(FPO/THF) was mainly synthesized by 2,2,3,3-tetrafluoropropyl ether (FPO) and tetrahydrofuran (THF) via cationic open-ring collection. (???)

The used chemical names are also not correct.

See Materials dimethy iolpropionic acid (DMPA)- ???

Results

Page 7 TGA all samples degraded completely at 450 ℃. How it possible if the sample contains 5 wt% of POSS Should stay Si containing ash.

In this form the paper should be reject.

Response: Thanks for the comment. I have corrected the full manuscript, thank you very much.

Reviewer 3 Report

The paper concerns preparation of fluorine-containing polyether waterborne polyurethane with POSS as cross-linking agent. The work is related to rapidly developing polymeric materials such as polyurethane materials. However, it contains some arguable elements and experimental shortcomings. A proper explanations are needed. Comments and reservations:

  • 1. Introduction - was written very generally. There is no concise literature review focusing on the analyzed topic. It is worth emphasizing the scientific novelty.
  • What kind of polyurethane can be included? Coatings, adhesives, etc.? What is the purpose of the received material? Please add specific examples of application potential.
  • 2.1. Materials - Why was its aliphatic form (isophorone diisocyanate - IPDI) used as the isocyanate? Why was no prepolymer used? Why was triethylamine (TEA) used and for what purpose? What was the criterion for selecting the dibutyltin dilaurate (DBTDL) catalyst? It is a very reactive organometallic (tin) catalyst. Why were cheaper and less toxic catalysts, e.g. amine, not used?
  • 2.4. Preparation of POSS-based FWPU dispersions - What was the assumed isocyanate index of the obtained polyurethane?
  • 2.5. Preparation of POSS-based FWPU films - Why is it taking so long to receive POSS-based FWPU films. Three days is a process that is definitely too long, which excludes its application potential. It is puzzling that despite the use of a strong catalyst, this cross-linking time is so long. Please explain.
  • Figure 2 - The numerical values of the wavenumber are not visible. How will the Authors explain the lack of characteristic bands at the wavelength of approx. 2275 cm-1 coming from the isocyanate group (N=C=O) for the tested samples? Typically, small amounts of isocyanate groups remain unreacted.
  • 3.3. Water contact angle (WCA) and mechanical properties of the different films - What standard were the mechanical tests based on? This is the key information to test the mechanical properties of the obtained polymer.

In the case of the FWPU1 sample, tearing (material damage) is noticeable for the value of 515 MPa and the analysis should be completed at this value.

Did the authors perform other key mechanical tests such as abrasion resistance, hardness or tear strength?

·        4. Conclusion - Very generally written. Please complete with specific result values.

  • There is a few of editorial and grammar mistakes, so paper needs to be improved carefully once again before another consideration process.

Final remark: In my opinion, the paper is worth recommendation for publication in Polymers after major revision. 

Author Response

Point-to-point responses to the comments from the reviewers

We would like to express our sincere appreciation for the useful comments and constructive suggestions from the reviewer 3. The resubmitted manuscript has been completely revised according to all the comments. The itemized responses to each the comments are listed as below.

The paper concerns preparation of fluorine-containing polyether waterborne polyurethane with POSS as cross-linking agent. The work is related to rapidly developing polymeric materials such as polyurethane materials. However, it contains some arguable elements and experimental shortcomings. A proper explanations are needed. Comments and reservations:

  • Point 1: Introduction - was written very generally. There is no concise literature review focusing on the analyzed topic. It is worth emphasizing the scientific novelty.

Response 1: Thanks for the comment. I have modified the introduction.

  • Point 2: What kind of polyurethane can be included? Coatings, adhesives, etc.? What is the purpose of the received material? Please add specific examples of application potential.

Response 2: Thanks for the comment. Waterborne polyurethane is an environmentally friendly material. It has excellent thermodynamic properties and can be used in coating, packaging and ink fields.

  • Point 3: 2.1. Materials - Why was its aliphatic form (isophorone diisocyanate - IPDI) used as the isocyanate? Why was no prepolymer used? Why was triethylamine (TEA) used and for what purpose? What was the criterion for selecting the dibutyltin dilaurate (DBTDL) catalyst? It is a very reactive organometallic (tin) catalyst. Why were cheaper and less toxic catalysts, e.g. amine, not used?

Response 3: Thanks for the comment. Compared with the isocyanate TDI of aromatic ring, IPDI is more stable. This reaction was followed by the prepolymer method. Detailed scheme is shown in 2.4 Preparation of POSS-based FWPU dispersions. In addition, for the synthesis of waterborne polyurethane, catalyst DBTDL is the most common and most used. DBTDL catalyst is very efficient and used in very small quantities (as a catalyst for 1% of the total mass).

  • Point 4: 2.4. Preparation of POSS-based FWPU dispersions - What was the assumed isocyanate index of the obtained polyurethane?

Response 4: Thanks for the comment. My work used hydroxyl modified POSS crosslinked and terminated polyurethanes to ultimately form FWPU. Studies such as: http://dx.doi.org/10.1016/j.polymer.2016.09.034

  • Point 5: Preparation of POSS-based FWPU films - Why is it taking so long to receive POSS-based FWPU films. Three days is a process that is definitely too long, which excludes its application potential. It is puzzling that despite the use of a strong catalyst, this crosslinking time is so long. Please explain.

Response 5: Thanks for the comment. Because of the water based polyurethane emulsion, and then I want it to evaporate a lot of water at room temperature to form a film. The dry membrane is then placed in an oven to remove the remaining water. This operation in order to prevent rapid drying caused by uneven structure of the material, resulting that the film character cannot show the best. In fact, the internal crosslinking has been completed, and the subsequent film forming step is the process to remove the aqueous solution to form a film.

  • Point 6: Figure 2 - The numerical values of the wavenumber are not visible. How will the Authors explain the lack of characteristic bands at the wavelength of approx. 2275 cm-1 coming from the isocyanate group (N=C=O) for the tested samples? Typically, small amounts of isocyanate groups remain unreacted.

Response 6: Thanks for the comment. My work used hydroxyl modified POSS crosslinked and terminated polyurethanes to ultimately form FWPU. So, all the isocyanate is consumed, and there is no peak at 2275 cm-1. Studies such as: http://dx.doi.org/10.1016/j.polymer.2016.09.034.

  • Point 7: 3.3. Water contact angle (WCA) and mechanical properties of the different films - What standard were the mechanical tests based on? This is the key information to test the mechanical properties of the obtained polymer.

In the case of the FWPU1 sample, tearing (material damage) is noticeable for the value of 515 MPa and the analysis should be completed at this value.

Did the authors perform other key mechanical tests such as abrasion resistance, hardness or tear strength?

Response 7: Thanks for the comment. I have supplemented the test criteria in the manuscript. The significant material damage of FWPU1 at 515% elongation indicated that the membrane material had reached the critical value. For this work. We would study the synthesis strategy of materials and explore the influence of different contents of POSS crosslinkers on the overall mechanical effect of materials. Finally, the mechanical effect of the material was the excellent when the POSS content reached 1%. The abrasion resistance, hardness or tear strength of the material was not further explored. Thank you very much for your comments.

  • Point 8: Conclusion - Very generally written. Please complete with specific result values. There is a few of editorial and grammar mistakes, so paper needs to be improved carefully once again before another consideration process.

Response 8: Thanks for the comment. I have revised the full manuscript.

Final remark: In my opinion, the paper is worth recommendation for publication in Polymers after major revision.

Reviewer 4 Report

Dear Sir,

The authors’ approach is interesting, but the manuscript suffers from numerous setbacks, one of the most important being the poor quality of the language used. I will not make an extensive list of all grammar mistakes, orthography, awkward phrasing etc., but will give only one example: starting from the first sentence of the abstract, the phrasing is wrong. “While waterborne polyurethane has optimized low volatile organic compounds (VOCs) and eco-friendly materials, the abundant of hydrophilic groups have not yet reached good mechanical properties, durability and hydrophobicity behaviors.” – “Waterborne polyurethane are more eco-friendly materials due to a lower VOCs (mainly isocyanates) content than the alternative materials. However, these rich hydrophilic groups polymers have not yet reached good mechanical properties, durability and hydrophobicity behaviors.” Some chemical names are not correct according to IUPAC: the compound the authors have named 2,2,3,3-tetrafluoropropyl ether (FPO) is in fact glycidyl 2,2,3,3-tetrafluoro-propyl ether, or more correctly named 2-(2,2,3,3-tetrafluoro-propoxymethyl)-oxirane (thus, the abbreviation FPO). Using the correct name is paramount in this case, since the compound participates to the polymerization process through its epoxy ring. Another example is dimethy lolpropionic acid (DMPA), named also  dimethy iolpropionic acid (DMPA), which is in fact dimethylolpropionic acid.

Thus, one of the first queries for this manuscript is a complete correction of the language used and a verification for corectness of all chemical names.

The Introduction paragraph is rather short, failing to mention published results that are pertinent to the subject: the idea of combining fluorinated moieties with polysiloxane in FWPU is not a new one. No mention of this is made in the Introduction (see for example Sui et al., Preparation and properties of polysiloxane modified fluorine-containing waterborne polyurethane emulsion, Progress in Organic Coatings, 2022, 106783 etc.). Another important paper that should have been included in the Introduction rationale is ref. 34, used in a hardly understandable sentence due to an rather upsetting repetition (“improved to lead to the significant improvement”): “Thus, the interaction between molecules improved to lead to the significant improvement of the mechanical properties of the composite materials[34,35].” Ref. 34 is in fact Chen’s work entitled “Waterborne POSS-silane-urethane hybrid polymer and the fluorinated films”, being thus an early example POSS inclusion in FWPUs.

What would be the advantage of the authors’ polymer over the ones that uses fluorinated POSS (e.g. Kannan et al., Fluoro-silsesquioxane-urethane Hybrid for Thin Film Applications, ACS Appl. Mater. Interfaces 2009, 1(2), 336–347)? Compare the results between a FWPU in which fluorine are present on the POSS structure and the one in which fluorine is present in a separate moiety.

Stress and strain tests showed that the best results were for a 1% POSS content. Did the authors explore the possibility to test content values closer to 1% (0.75%, 1.1%, 1.25% and so on)?

 Other comments:

- “The characteristic signals at 1234 cm-1 and 1535 cm−1 respectively cor-responded to the C=O of the polycarbonate glycol and N-H bond in the ethyl carbamate group.” ?? What “polycarbonate glycol” and what “ethyl carbamate group”? There is no carbonate moiety on the structure, and the C=O in urethanes is rather in the 1700s; it’s the bending of the C(O)-NH bond that appears around 1540. An etheric C-O-C could be the 1234 signal.

- if possible, in Fig 3 do not represent up to 800C, but limit to 500C and widen or expand on the horizontal the graphs in order to better see the differences between the 4 curves.

- How were the 4 polymers obtained exactly? Was it a single batch of the polyurethane obtained and next 3/4 of the mother solution separated in different vessels and POSS added in 1, 3 and 5%, or were there 4 different batches of the PU obtained in 4 different vessels? Because, if 4 different batches were prepared in 4 different flasks, what differences could there be between the ratio of glyceryl, tetrafluoromethyl and dimethylolpropionic acid in the PU polymer, previous to POSS inclusion? If there was only one batch of PU, then there would be no differences in PU used for POSS inclusion.

 Minor comments:

- 2-hydroxy-1-ethanethiol is mentioned twice in the list of chemicals purchased

- Ref. 12 is rather old (2002), it can be removed or replaced with a more recent one (e.g. Król, P., Pielichowska, K., Król, B. et al. Polyurethane cationomers containing fluorinated soft segments with hydrophobic properties. Colloid Polym Sci, 2021, 299, 1011–1029)

- numerous blank spaces where they should not be (e.g. “boron trifluoride di-ethy lether” etc.)

- “thermal analysis of four polymers” – which four polymers? Those from Table 1?

- precise in the legends of Fig 6 and 7 that figures a to d corespond to FWPUs 0, 1, 3 and 5

 Overall, the manuscript is interesting, but needs major changes before acceptance, as follows:

-          Complete re-editing of the text for a comprehensive revision of the language used, correct chemical nomenclature

-          Complement of information regarding the following aspects:

1)      PU with POSS – what POSS have brought over simple PU

2)      PU with F – what the presence of fluorine has brought in the PU

3)      PU with POSS and F, in which F is in the POSS

4)      PU with POSS and F, in which F is outside the POSS

Author Response

Point-to-point responses to the comments from the reviewers

We would like to express our sincere appreciation for the useful comments and constructive suggestions from the reviewer 1. The resubmitted manuscript has been completely revised according to all the comments. The itemized responses to each the comments are listed as below.

Point 1: The authors’ approach is interesting, but the manuscript suffers from numerous setbacks, one of the most important being the poor quality of the language used. I will not make an extensive list of all grammar mistakes, orthography, awkward phrasing etc., but will give only one example: starting from the first sentence of the abstract, the phrasing is wrong. “While waterborne polyurethane has optimized low volatile organic compounds (VOCs) and eco-friendly materials, the abundant of hydrophilic groups have not yet reached good mechanical properties, durability and hydrophobicity behaviors.” – “Waterborne polyurethane are more eco-friendly materials due to a lower VOCs (mainly isocyanates) content than the alternative materials. However, these rich hydrophilic groups polymers have not yet reached good mechanical properties, durability and hydrophobicity behaviors.” Some chemical names are not correct according to IUPAC: the compound the authors have named 2,2,3,3-tetrafluoropropyl ether (FPO) is in fact glycidyl 2,2,3,3-tetrafluoro-propyl ether, or more correctly named 2-(2,2,3,3-tetrafluoro-propoxymethyl)-oxirane (thus, the abbreviation FPO). Using the correct name is paramount in this case, since the compound participates to the polymerization process through its epoxy ring. Another example is dimethy lolpropionic acid (DMPA), named also dimethyl iolpropionic acid (DMPA), which is in fact dimethylolpropionic acid.

Thus, one of the first queries for this manuscript is a complete correction of the language used and a verification for corectness of all chemical names.

Response 1: Thanks for the comment. I have corrected it in the manuscript. Thank you very much for your very careful suggestions on this manuscript.

Point 2: The Introduction paragraph is rather short, failing to mention published results that are pertinent to the subject: the idea of combining fluorinated moieties with polysiloxane in FWPU is not a new one. No mention of this is made in the Introduction (see for example Sui et al., Preparation and properties of polysiloxane modified fluorine-containing waterborne polyurethane emulsion, Progress in Organic Coatings, 2022, 106783 etc.). Another important paper that should have been included in the Introduction rationale is ref. 34, used in a hardly understandable sentence due to an rather upsetting repetition (“improved to lead to the significant improvement”): “Thus, the interaction between molecules improved to lead to the significant improvement of the mechanical properties of the composite materials[34,35].” Ref. 34 is in fact Chen’s work entitled “Waterborne POSS-silane-urethane hybrid polymer and the fluorinated films”, being thus an early example POSS inclusion in FWPUs.

Response 2: Thanks for the comment. I have supplemented the reference in the introduction.

Point 3: What would be the advantage of the authors’ polymer over the ones that uses fluorinated POSS (e.g. Kannan et al., Fluoro-silsesquioxane-urethane Hybrid for Thin Film Applications, ACS Appl. Mater. Interfaces 2009, 1(2), 336–347)? Compare the results between a FWPU in which fluorine are present on the POSS structure and the one in which fluorine is present in a separate moiety.

Response 3: Thanks for the comment. We are sure that the hydrophobic effect of grafting fluorine units on POSS and then grafting into waterborne polyurethane is definitely better. But it would take more steps and cost a lot more. In this study, cationic ring-opening polymerization was used to obtain fluorinated polyether polymers. This one - step synthesis can be obtained fluorinated polyether, and the method is very simple. Then the polyether has good flexibility, so that the whole polymer has a lower glass transition temperature. In addition, the rigid POSS structure is introduced into the hard chain segment, and the flexibility of polyurethane can be further compensated by polyether fluoropolymer. Just as the reviewer pointed out in the article (e.g. Kannan et al., Fluoro-silsesquioxane-urethane Hybrid for Thin Film Applications, ACS Appl. Mater. Interfaces 2009, 1(2), 336–347), small molecule fluorine monomer is connected to waterborne polyurethane, which is used as soft chain segment, and the hydrophobicity definitely greatly improved. But this will greatly reduce the function of polyurethane soft chain segment. This will greatly reduce the flexibility of polyurethanes when the soft chain segments are polyether polymers. Thanks again for your very valuable comments.

Point 4: Stress and strain tests showed that the best results were for a 1% POSS content. Did the authors explore the possibility to test content values closer to 1% (0.75%, 1.1%, 1.25% and so on)?

Response 4: Thanks for the comment. In this work, we mainly discuss the strategy of combining our new fluorinated monomers with POSS in waterborne polyurethane. Then we came up with a few ratios to explore their performance and finally came to a conclusion. As the reviewer said, we can do some experiments closer to 1%. This way we can get better application results. Thank you very much for your advice. But this optimal material effect is not what we are looking for in this experiment. We discuss the application of our newly synthesized fluoropolymer and POSS to synthesize waterborne polyurethane.

Point 5: Other comments:

- “The characteristic signals at 1234 cm-1 and 1535 cm−1 respectively cor-responded to the C=O of the polycarbonate glycol and N-H bond in the ethyl carbamate group.”?? What “polycarbonate glycol” and what “ethyl carbamate group”? There is no carbonate moiety on the structure, and the C=O in urethanes is rather in the 1700s; it’s the bending of the C(O)-NH bond that appears around 1540. An etheric C-O-C could be the 1234 signal.

Response 5: Thanks for the comment. I have corrected it, thank you very much.

Point 6: - if possible, in Fig 3 do not represent up to 800C, but limit to 500C and widen or expand on the horizontal the graphs in order to better see the differences between the 4 curves.

Response 6: Thanks for the comment. I have corrected the Fig 3.

Point 7: - How were the 4 polymers obtained exactly? Was it a single batch of the polyurethane obtained and next 3/4 of the mother solution separated in different vessels and POSS added in 1, 3 and 5%, or were there 4 different batches of the PU obtained in 4 different vessels? Because, if 4 different batches were prepared in 4 different flasks, what differences could there be between the ratio of glyceryl, tetrafluoromethyl and dimethylolpropionic acid in the PU polymer, previous to POSS inclusion? If there was only one batch of PU, then there would be no differences in PU used for POSS inclusion.

Response 7: Thanks for the comment. I synthesized these four kinds of waterborne polyurethanes according to the prepolymer method. Firstly, I synthesized IPDI, P(FPO/THF) and DMPA in a certain proportion to obtain the prepolymer. Then different proportions of POSS are added to the prepolymer structure to form the final waterborne polyurethane WPU. Other reaction conditions were controlled the same, so only the properties of water-based polyurethane with different POSS contents were compared. The detailed reaction process is in 2.4 Preparation of POSS-based FWPU dispersions.

Point 8: Minor comments:

- 2-hydroxy-1-ethanethiol is mentioned twice in the list of chemicals purchased

Response 8: Thanks for the comment. I've modified it in the manuscript.

Point 9: - Ref. 12 is rather old (2002), it can be removed or replaced with a more recent one (e.g. Król, P., Pielichowska, K., Król, B. et al. Polyurethane cationomers containing fluorinated soft segments with hydrophobic properties. Colloid Polym Sci, 2021, 299, 1011–1029)

Response 9: Thanks for the comment. The reference had been added.

Point 10: - numerous blank spaces where they should not be (e.g. “boron trifluoride diethy lether” etc.)

Response 10: Thanks for the comment. The mistakes had been revised.

Point 11: - “thermal analysis of four polymers” – which four polymers? Those from Table 1?

Response 11: Thanks for the comment. I have revised it in the manuscript。

Point 12: - precise in the legends of Fig 6 and 7 that figures a to d corespond to FWPUs 0, 1, 3 and 5

Response 12: Thanks for the comment. I have refined Figures 6 and 7.

Point 13: Overall, the manuscript is interesting, but needs major changes before acceptance, as follows:

- Complete re-editing of the text for a comprehensive revision of the language used, correct chemical nomenclature

- Complement of information regarding the following aspects:

1) PU with POSS – what POSS have brought over simple PU

2) PU with F – what the presence of fluorine has brought in the PU

3) PU with POSS and F, in which F is in the POSS

4) PU with POSS and F, in which F is outside the POSS

Response 13: Thanks for the comment. I have revised the whole manuscript. Many thanks to the reviewers for their detailed review and suggestions to make the manuscript more perfect. Thanks again.

Round 2

Reviewer 1 Report

The authors failed to fully address my comments; therefore, the manuscript cannot be accepted in the current format. Please fully address the comments below:

1- The authors did not discuss whether dynamic CA measurements (change of CA with time) are needed to better explore the fluorine-containing polyether waterborne polyurethane with POSS as the crosslinking agent.

2- Scale bars are still missing from Figure 4 (I don't accept the justifications made by the authors). 

3- The limitations of the optical contact angle measuring instrument (OCA15EC, Germany Defi Instrument Co., Ltd.) via the sessile method were not appreciated, see my old comments and explain how limitations of polynomial fitting can be addressed by newer techniques.

4- The paper is still filled with grammatical mistakes. For example:

·      “The polyhedral oligomeric silsesquioxane (POSS) depends on the special structure to be the optimal choices.” The word “choices” should be “choice

·      “As shown in Figure 3 (a), the four films decomposed at temperature between 200-450 ℃ “ The word “temperature” should be plural. 

Reviewer 2 Report

The second version of paper Deng et al. stay have many disadvantages.

First of all stay a problem with English. The author used probably the PC software translation which leads to hard-to-read sentences.  

There are false statements in the interpretation of the results.

1.  Abstract

isoflurane diisocyanate (IPDI) is really Isophorone diisocyanate

triethylamine (TEA) were used as hydrophilic unit

no, it used as catalyst

the glass transition temperature could reach about at -50 ℃ identifying the good mechanical properties even at the low temperature.

False, connection between Tg and mechanical performance is absent

Figure 1. Authors wrote (see Exp) that NMR spectra were recorded in chloroform=d solution. But as can see from Figure 1a this spectrum was recorded in DMSO-d6 (present the rest signals of solvent). As a result, exist misinterpretation. The signal a is not the signal of OH group (it fall in water signal in DMSO at about 3.3. ppm)

Page 8 all samples degraded completely at 450 ℃.

No, as one can see (Figure 3f) the POSS units stay

All interpretation of thermal degradation is false and speculative

This was because POSS contains a large number of rigid silicon-oxygen bonds, which reduced the mobility of the molecular chains of waterborne polyurethane.

The silicon-oxygen bonds are flexible. The decrease in molecular mobility has a different nature.  

The paper should be major revised before publication

Reviewer 4 Report

Dear Sir,

 The authors have satisfactorily answered to all comments.

However, the quality of the language used has only slightly being improved.

Another comment is regarding the real role played by dimethylolpropionic acid (DMPA) and triethylamine (TEA) – the authors assessed in the Absctract that these compounds were used as hydrophilic unit (!!?). I think that they can be considered at least catalysts, if not even reagents (since in praragraph 2.4. the authors mentioned that “TEA was added and reacted with neutralize carboxyl group of DMPA with 1:1 M ratio” – this expression is also an example of the fact that the text was not re-edited for correct Engliah language) – by the way, the quantities in which these compounds were added should have been mentioned in pargraph 2.4.

Another concern is the consideration that POSS is a “hollow rigid cage”- the authors have based some of their explanation on such a rigid structure. The assessment should have been based on some previous consideration on the flexibility/rigidity of such structures (silsesiloxanes) – see for example Early, A Quantum Mechanical Investigation of Silsesquioxane Cages, J. Phys. Chem., 1994, 98, 8693-8698 (who considered them rigid) or Neyertz et al., The structure of amino-functionalized polyhedral oligomeric silsesquioxanes (POSS) studied by molecular dynamics simulations, Computational Materials Science, 2012, 62, 258-265 (who considered them fairly rigid).

Thus, the manuscript is still not ready for acceptance.

Round 3

Reviewer 1 Report

Comments are addressed. 

Author Response

Thank you for the comments on this manuscript.

Reviewer 2 Report

3th reviewer comments on paper Ding et al.

First of all I want to ask the authors: are you read their paper?

1.       Abstract

Line 25 ‘the glass transition temperature could reach about at -50 .’

Page 8 line 245 ‘soft chain segment in the FWPU network induced 244 the Tg value of all the FWPU films about at 50 , as shown in Figure 3(c).’

2.       Line 245-247 “in the composite FWPU films, the more content of POSS added, the higher Tm value in- 246 creased. For example, the FWPU0 film showed the lowest Tg at 92.6 , while the FWPU5 247 exhibited the highest Tg at 112.3 .”

What you described- melting or glass transition?

3.       Please improve your English

Author Response

We would like to express our sincere appreciation for the useful comments and constructive suggestions from the reviewers. The resubmitted manuscript has been completely revised according to all the comments. The itemized responses to each the comments are listed as below.

Point 1: Abstract
Line 25 ‘the glass transition temperature could reach about at -50 ℃.’

Page 8 line 245 ‘soft chain segment in the FWPU network induced 244 the Tg value of all the FWPU films about at 50 ℃, as shown in Figure 3(c).’

Response 1: Thanks for the comment. I have modified it in the page 8 line 245.

Point 2: Line 245-247 “in the composite FWPU films, the more content of POSS added, the higher Tm value in- 246 creased. For example, the FWPU0 film showed the lowest Tg at 92.6 ℃, while the FWPU5 247 exhibited the highest Tg at 112.3 ℃.”

What you described- melting or glass transition?

Response 2: Thanks for the comment. I had been described the melting temperature and I have corrected this part. Thank you again for the correction.

Point 3: Please improve your English

Response 3: Thanks for the comment. I had revised the English of the whole manuscript.

Reviewer 4 Report

Dear Sir,

The authors have answered my comments and the manuscript has been re-edited for correct English language. Therefore, the manuscript could be considered for publication.

Author Response

(The authors gave the same response as above.)
